# Human Papillomavirus 16 E7 Promotes EGFR/PI3K/AKT1/NRF2 Signaling Pathway Contributing to PIR/NF-κB Activation in Oral Cancer Cells

**DOI:** 10.3390/cancers12071904

**Published:** 2020-07-15

**Authors:** Diego Carrillo-Beltrán, Juan P. Muñoz, Nahir Guerrero-Vásquez, Rancés Blanco, Oscar León, Vanesca de Souza Lino, Julio C. Tapia, Edio Maldonado, Karen Dubois-Camacho, Marcela A. Hermoso, Alejandro H. Corvalán, Gloria M. Calaf, Enrique Boccardo, Francisco Aguayo

**Affiliations:** 1Laboratorio de Oncovirología, Programa de Virología, Instituto de Ciencias Biomédicas (ICBM), Facultad de Medicina, Universidad de Chile, Santiago 8380000, Chile; diegocb17@hotmail.com (D.C.-B.); nahir.alejandra@hotmail.com (N.G.-V.); rancesblanco1976@gmail.com (R.B.); 2Instituto de Alta Investigación, Universidad de Tarapaca, Arica 1000000, Chile; jp_182_mb@hotmail.com (J.P.M.); gmc24@cumc.columbia.edu (G.M.C.); 3Departamento de Acuicultura y Recursos Agroalimentarios, Universidad de Los Lagos, Osorno 933, Chile; oscar.leon@uchile.cl; 4Department of Microbiology, Institute of Biomedical Sciences, University of Sao Paulo, Sao Paulo 05508-900, Brazil; vanesca_lino@hotmail.com; 5Programa Biología Celular y Molecular, Instituto de Ciencias Biomédicas (ICBM), Facultad de Medicina, Universidad de Chile, Santiago 8380000, Chile; jtapia@med.uchile.cl (J.C.T.); emaldona@med.uchile.cl (E.M.); 6Innate Immunity Laboratory, Immunology Program, Instituto de Ciencias biomédicas, Facultad de Medicina, Universidad de Chile, Santiago 8380000, Chile; kdubois@gmail.com (K.D.-C.); mhermoso@med.uchile.cl (M.A.H.); 7Hematology and Oncology Department, School of Medicine, Pontificia Universidad Católica de Chile, Santiago 8330024, Chile; acorvalan@uc.cl; 8Advanced Center for Chronic Diseases (ACCDiS), Pontificia Universidad Católica de Chile, Santiago 8330024, Chile; 9Center for Radiological Research, Columbia University Medical Center, New York, NY 10032, USA

**Keywords:** papillomavirus, pirin, NF-κB, oncoprotein, oral, cancer

## Abstract

A subset of oral carcinomas is etiologically related to high-risk human papillomavirus (HR-HPV) infection, with HPV16 being the most frequent HR-HPV type found in these carcinomas. The oncogenic role of HR-HPV is strongly dependent on the overexpression of E6 and E7 oncoproteins, which, in turn, induce p53 and pRb degradation, respectively. Additionally, it has been suggested that HR-HPV oncoproteins are involved in the regulation of nuclear factor kappa-light-chain-enhancer of activated B cells (NF-κB), inducing cancer progression and metastasis. Previously, we reported that HPV16 E7 oncoprotein promotes Pirin upregulation resulting in increased epithelial–mesenchymal transition (EMT) and cell migration, with Pirin being an oxidative stress sensor and activator of NF-κB. In this study, we demonstrate the mechanism by which HPV16 E7-mediated Pirin overexpression occurs by promoting EGFR/PI3K/AKT1/NRF2 signaling, thus causing PIR/NF-κB activation in oral tumor cells. Our results demonstrate a new mechanism by which E7 contributes to oral cancer progression, proposing PIR as a potential new therapeutic target.

## 1. Introduction

Human papillomavirus (HPV) is a double-stranded DNA virus with approximately 8000 base pairs and exclusive epithelial tropism. HPV is the most common sexual infection found in the general population, affecting both women and men. Mucosal HPV types are classified according to their capacity to induce malignant transformation in high-risk HPV (HR-HPV) or low-risk HPV (LR-HPV). HR-HPVs are involved in the development of almost 100% of cervical carcinomas worldwide [1,2]. Additionally, these viruses are etiologically involved in the development of other anogenital tumors and head and neck cancers (HNCs), including oropharyngeal (OPCs) and oral carcinomas (OC) [3,4,5] with HPV16 being the most prevalent in HPV-associated carcinomas [6].

It is known that HR-HPV oncogenic potential is related to the overexpression of E6 and E7 proteins, which in turn, mediate the degradation of p53 and pRb tumor suppressor proteins, respectively [7,8]. However, a plethora of E6 and E7 activities independent of p53 and pRb have been described [9], one example being that both E6 and E7 are involved in chronic oxidative stress (OS), DNA damage and genomic instability in HNC cells [10]. Particularly, HR-HPV E7 accelerates the turnover of claspin, which is involved in the regulation of DNA damage signaling [11]; other partners of E7 include p21 [12] and p27 [13], participating in cell cycle arrest during epithelial differentiation. E7 also interacts with the so-called “pocket proteins”, such as p107 and p130, which positively regulate the cell cycle by joining to E2F4 and E2F5 proteins, respectively [14]. Additionally, HPV16 E7 has been involved in the regulation of EGFR/PI3K/AKT signaling pathway [15,16], controlling cell cycle, survival, metabolism, motility, and genomic instability [17]. In this respect, it has been reported that HPV16 E7 promotes cell migration and metastasis in an AKT-dependent manner [18]. Notably, it is estimated that approximately 50% of patients with oral squamous cell carcinoma (OSCC), the most common histological type of OC, show lymph node metastasis at the moment of clinical diagnosis [19,20]. However, it remains unclear how HR-HPV is involved in OSCC progression and metastasis, with limited related studies regarding HR-HPV tumor progression in this anatomical region.

Previously, we reported that HPV16 E7 is involved in Pirin (encoded by *PIR* gene) overexpression, which is an OS sensor and activator of nuclear factor kappa-light-chain-enhancer of activated B cells (NF-κB) [21]. Upon NF-κB activation, Pirin increases epithelial–mesenchymal transition (EMT) and cell migration in HeLa, a HR-HPV positive cell line [22]. The NF-κB pathway is composed of homo or heterodimers of five proteins belonging to the REL oncogene family, these proteins being p50 (NF-κB1), p52 (NF-κB2), p65 (Rel A), Rel B and c-Rel [23]. Accordingly, the NF-κB pathway is recognized by its key role in inflammation and innate immune response, plus, it is related with tumor progression and increased cell migration [24]. However, conflicting data are found regarding the role of HR-HPV and viral oncogenes in NF-κB activation. Moreover, although factors involved in a such differences are unclear [25,26,27], it seems that NF-κB is activated in a cell type-dependent manner [28]. Here, we addressed the role of signaling pathways involved in HPV16 E7-mediated PIR/NF-κB activation and oral cell migration, finding that HPV16 E7 promotes the activation of the EGFR/PI3K/AKT/NRF2 signaling pathway, in turn stimulating PIR-mediated NF-κB activation in oral cancer cells.

## 2. Results

### 2.1. HPV16 E7 Oncoprotein Upregulates the Levels of Pirin in Oral Cells

Floor of mouth squamous cell carcinoma (SCC143) cells were transduced with HPV16 pLXSNE7 or pLXSN (empty) vector. Cell colonies were pooled and named SCC143/E7 and SCC143/V, respectively. The levels of E7 transcripts and protein were evaluated by RT-PCR and Western blot, respectively. As expected, E7 transcripts and protein were detected in SCC143/E7 cells and were not detected in SCC143/V cells (Appendix A). In addition, E7 protein was capable of promoting pRb downregulation and cell proliferation, demonstrating the functional activity of this oncoprotein. Moreover, amphiregulin (AREG) upregulation by E7 was confirmed, as previously reported [29] (Appendix A).

We observed that PIR at mRNA and protein levels were significantly increased in SCC143/E7 cells compared with control cells, as shown in Figure 1A,C. In addition, E7 knockdown by siRNA showed a significant decrease in Pirin levels, demonstrated by immunofluorescence in SCC143/E7 cells (Figure 1D). The functionality of siRNA for PIR or E7 knockdown is shown Appendix A. Next, we decided to analyze the behavior of Pirin in the presence of ectopic E7 expression in a more physiological context, consisting of stratified epithelia. Therefore, we confirmed that Pirin was positively regulated in organotypic raft cultures established from oral keratinocyte of floor of mouth (OKF6-TERT2) cells transduced with HPV16 pLXSNE7 (Figure 1E,F). In addition, the functionality of E7 was confirmed by pRb downregulation in the rafts (Appendix A). Moreover, the same Pirin response was observed in organotypic raft cultures established from human foreskin keratinocytes (HFK) overexpressing HPV16 E7. In addition, Pirin was upregulated to a lesser extent in organotypic cultures established from cells that expressed the E7Δ21-24 mutant, revealing that the pRb-binding site is important for E7-mediated PIR upregulation (Figure 1G). Taken together, these data strongly suggest that HPV16 E7 promotes an increase in PIR transcripts and Pirin levels in oral epithelial cells. Moreover, they show that this effect is associated, at least in part, with the integrity of sequences in E7 required to induce pRb degradation.

### 2.2. HPV16 E7 Promotes NF-κB Activation

In order to evaluate NF-κB activation by E7, the pHAGE/NF-κB vector (Figure 2A) containing consensus NF-κB response elements was transfected in both SCC143/E7 and SCC143/V cells. We observed a significant increase in luciferase activity, normalized against green fluorescent protein (GFP) in SCC143/E7 cells when compared with control cells (Figure 2B). To confirm the role of E7 oncoprotein and Pirin in NF-κB activation, SCC143/E7 cells were co-transfected with both pHAGE/NF-κB vector and specific siRNAs for E7 or PIR knockdown. We observed a significant and similar decrease in NF-κB activation when both E7 or PIR siRNAs were used, suggesting that both E7 and Pirin are involved in NF-κB activation (Figure 2C). To determine which proteins are involved in NF-κB signaling pathway activation in the presence of E7, we used a proteomic approach (Appendix A), observing that SCC143/E7 cells strongly expressed NF-κB-associated proteins. IRF5, IRF8, p-p65^S529^, c-Rel and IκB among others were positively regulated in the presence of E7. On the contrary, JNK1/2, IL-18Ra, CD40 and NF-κB2 (the non-canonical NF-κB pathway) among others, were negatively regulated (Figure 2D and Appendix A). For further NF-κB characterization, the specific activation of NF-κB proteins in SCC143/E7 cells was evaluated by Western blot, where, in the presence of E7, an increase in nuclear levels of c-Rel and p65^S529^ (Figure 2E) and a decrease in cytoplasmic and nuclear levels of p65^S536^ was observed (Appendix A). Taken together, these data suggest that HPV16 E7 promotes a Pirin-dependent canonical NF-κB activation by a mechanism mediated by c-Rel and p65^S529^.

### 2.3. HPV16 E7 Induces EGFR/PI3K/AKT1 Signaling for PIR/NF-κB Activation

To further investigate the signaling pathways involved in Pirin-dependent NF-κB activation by E7, we decided to analyze the activation state of MAPK and PI3K pathway components in cells expressing HPV16 E7. For this purpose, we used a phosphoproteomic approach to evaluate 27 phosphorylated MAPK and PI3K/AKT proteins (Appendix A). We observed that phosphorylated proteins positively regulated in the presence of HPV16 E7 were AKT1, P70, GSK3B, TOR, p38a, p38b, RSK3, p53, RSK2, CREB and GSK3A. Alternatively, the phosphorylated form of JNK, ERK and p38g proteins were negatively regulated (Figure 3A and Appendix A). The observed increase in some specific phosphorylated proteins was confirmed by Western blot (Appendix A). In addition, we detected increased levels of total and phosphorylated EGFR upon ectopic E7 expression (Figure 3B). In organotypic raft cultures established from OKF6-TERT2 oral cells, we confirmed that E7 expression was associated with increased EGFR and AKT1^S473^ proteins levels (Figure 3C,D). A similar increase in AKT1^S473^ levels was observed in organotypic cultures of PHK expressing wild type E7. However, this was not observed in rafts established from PHK transduced with E7Δ21-24 mutant, demonstrating the importance of the pRb-binding site in the regulation of AKT1 mediated by E7 (Appendix A). Taken together, these results suggest that the EGFR/PI3K/AKT1 pathway is activated in the presence of HPV16 E7. In order to determine the role of this pathway during E7-mediated PIR/NF-κB activation, pharmacologic inhibitors were used. Therefore, we used Gefitinib (EGFR inhibitor), LY294002 (PI3K inhibitor) and U0126 (ERK inhibitor) for 24 h. Although inhibition of EGFR, PI3K and ERK promoted a decrease in Pirin levels, the more significant decrease occurred when PI3K and ERK were inhibited (Figure 4A). It should be noted that an indirect inhibition of AKT1 was detected by using the U0126 inhibitor. To address the role of EGFR during PIR signaling, a time–response assay ranging between 0 and 24 h was carried out after Gefitinib exposure (Figure 4B). A significant decrease in Pirin levels after 3 h of Gefitinib exposure was observed. In order to confirm the role of PI3K/AKT1 pathway, a time–response assay between 0 and 24 h was carried out after LY294002 exposure. Under these conditions, we found a significant decrease in Pirin levels after 12 h of LY294002 exposure (Figure 4C). To evaluate the EGFR/PI3K/AKT1 signaling and PIR/NF-κB activation by E7 in organotypic rafts, we used Gefitinib for 3 h and LY294002 for 12/24 h. We found a significant decrease in Pirin levels with both inhibitors in OKF6-TERT2 E7 organotypic rafts (Figure 4D,E). In addition, we observed a decrease in Pirin levels in PHK rafts treated with LY294002 at 24 h (Appendix A). Finally, we observed a significant decrease in c-Rel and p65^S529^ levels and NF-κB activation when using the inhibitor LY294002 for 12/24 h in organotypic OKF6/TERT2 E7 and SCC143/E7 cells, suggesting that PI3K/AKT signaling is involved in E7-induced NF-κB activation (Figure 4F,G). These results reveal that HPV16 E7 promotes EGFR activation and PI3K/AKT1 signaling, which, in turn, promotes an increase in Pirin levels and canonical NF-κB activation.

### 2.4. HPV16 E7 Activates PIR-Promoter through NRF2 Transcription Factor in Oral Cells

In order to evaluate the E7-dependant transcriptional activation of PIR, we developed a construct in which the PIR promoter (−341 nt to +559 nt) was inserted upstream of the firefly luciferase reporter gene (luc2) open reading frame (pmiR-GLO/pPIR). After cell transfection with the pmiR-GLO/pPIR construct, we observed that SCC143/E7 cells showed a significant increase in luciferase activity when compared with SCC143/V cells (Figure 5A). Moreover, by using an interfering RNA for E7 knockdown or LY294002 inhibitor for 12 h, we observed a significant decrease in luciferase activity, suggesting that E7 and PI3K/AKT are involved in transcriptional activation of the PIR gene promoter (Figure 5B,C). To determine relevant transcription factors potentially involved in the E7-mediated PIR promoter activation, four mutants in the PIR promoter region were synthesized by using the PROMO 3.0 software. Considering that the E2F family of factors are released by E7-induced degradation of pRb and NRF2 is involved in the regulation of PIR transcript levels [30], we constructed E2F1- and NRF2 (antioxidant response elements (ARE))-binding site mutants in the PIR promoter, as shown in Figure 5D. We verified that the ARE site allowing NRF2 binding is relevant for inducing E7-mediated PIR promoter activation (Figure 5D, right). Then, establishing, in a timely manner, that the NRF2 factor is responsible for increasing PIR promoter activity, a chromatin immunoprecipitation (ChIP) assay was performed. We analyzed the c-Jun and c-Fos proteins components of the AP1 transcription factor, since there is a 12-*O*-tetradecanoilphorbol-13-acetate (TPA) response element (TRE) site very close to the ARE site. In addition, AP1 is a recognized transcription factor potentially regulated by the PI3K/AKT pathway. Accordingly, ChIP test results showed that the recruitment of NRF2/RNA Pol II into the PIR promoter in the presence of E7 is increased approximately 3.5-fold in respect to the control. In addition, c-Jun and c-Fos were not significantly recruited into the PIR promoter in the presence of E7 (Figure 5E). Moreover, a decrease in the recruitment of NRF2/RNA Pol II in the PIR promoter was shown when the cells were incubated with LY294002 inhibitor for 24 h (Figure 5F). Finally, a significant increase in NRF2 levels was observed in OKF6-TERT2 and SCC143 cells expressing E7 (Figure 5G). Taken together, these data demonstrate that HPV16 E7 oncoprotein promotes NRF2 transcription factor recruitment to the ARE site into the PIR promoter in oral cancer cells.

### 2.5. HPV16 E7-Mediated Pirin Overexpression Positively Regulates Cell Migration and Epithelial–Mesenchymal Transition (EMT)

To evaluate the consequences of E7-mediated Pirin overexpression, we studied migration and EMT in oral cells, finding that HPV16 E7 expression significantly increased transmigration in SCC143 cells and OKF6-TERT2 (Figure 6A and Appendix A). Evaluating the role of E7, p65 and Pirin in increased cell migration, a transwell assay was carried out in SCC143/E7 and squamous cell carcinoma of the oral lateral tongue (SCC47) (HPV16-positive cell line) cells previously transfected with a siRNA for E7, p65 or PIR knockdown, with the functionality of corresponding siRNA shown in Appendix A. We observed a significant decrease in cell migration after E7, p65 and PIR knockdown when compared with those cells transfected with a scrambled sequence (Figure 6B,C and Appendix A). To further confirm the role of Pirin overexpression, SCC143 oral cells were transfected with pcDNA 3.1+C–eGFP and pcDNA 3.1+C–eGFP-PIR. We confirmed ectopic expression of eGFP and eGFP–Pirin protein by Western blot (Figure 6D). Moreover, endogenous Pirin levels were evaluated by Western blot, observing similar levels in both conditions (eGFP and eGFP–Pirin), indicating that there are no differences in basal protein expression (Appendix A). Consecutively, a significant increase in cell transmigrating cells was observed in the presence of Pirin overexpression related to the control (Figure 6E). Finally, levels of EMT biomarkers such as E-cadherin and N-cadherin were evaluated, observing a significant decrease in E-cadherin and a significant increase in N-cadherin proteins in cells overexpressing Pirin with respect to the control (Figure 6F). These results suggest that both E7 and PIR are involved in the increased transmigration ability and EMT of oral epithelial cells.

## 3. Discussion

It has been widely reported that HPV oncoproteins promote DNA damage and OS involved in cancer initiation and progression [31,32,33]. However, the role of E7 oncoprotein in oral cancer progression has not been sufficiently addressed, although it is critical for immortalization and cell transformation [34]. In fact, HR-HPV E7 promotes pRb degradation, an important event in virus-mediated carcinogenesis, which is remarkable as pRb knockout mice do not completely recapitulate HR-HPV E7-mediated alterations, suggesting additional E7 effects [35,36]. Additionally, we have reported that HPV16 E7 is involved in *PIR* positive regulation [21] and. Interestingly. the translated protein, Pirin, harbors an Fe^+2^-binding site functioning as an OS sensor in NF-κB activation and cancer progression, whose overexpression is mediated by factors promoting OS.

In the present study, we have addressed some mechanisms by which PIR/NF-κB is activated by HPV16 E7 oncoprotein in oral epithelial tumor cells, finding that it positively regulates the levels of *PIR* transcripts and Pirin in 2D and 3D oral cell cultures. These results confirm those previously reported by us in other tumor epithelial cell lines including cervical cancer [21]. Next, we identified signaling pathways positively regulating Pirin, finding HPV16 E7 and Pirin involvement in NF-κB activation, although controversy still exists regarding HPV oncoprotein effect on NF-κB activation [25,37]. The canonical NF-κB pathway is triggered by factors promoting IκB degradation by the NF-κB Essential Modulator (NEMO) and IκB kinases (IKK1 and IKK2), leading to the nuclear translocation of the p50/RelA heterodimer [38]. Alternatively, the RelB/p52 NF-κB complex is predominant in the non-canonical NF-κB pathway and depends on inducible p100 processing rather than IκBα degradation [39]. Prominent activators of NF-κB signaling are tumor necrosis factor receptor (TNFR) family members, lipopolysaccharides (LPS) and EGFR [40,41], in fact, EGFR and NF-κB are described as co-directional activation signaling pathways in some carcinomas, such as HNCs [42]. Once activated, NF-κB regulates the expression of genes involved in different cellular events, including EMT and cancer progression, a process that is strongly related to E-cadherin downregulation and N-cadherin upregulation [43,44,45]. Pirin is relevant during NF-κB signaling, working as transcriptional factor with Rel-A protein (p65) [46] and, importantly, Pirin has also been described as an EMT regulator by decreasing E-cadherin expression [22,47]. Data from the NF-κB array demonstrates that IRF5, IL17 Ra and IRF8 were the most positively regulated proteins in the presence of E7, all were involved in interferon response regulation, although possible interactions with HPV oncoproteins have not been reported. Furthermore, we found that C-Rel, IκB and p65^S529^, among others, are positively regulated by HPV16 E7. Additionally, it is understood that HPV16 E7 positively regulates EGFR at RNA and protein level in foreskin keratinocytes [15] and, interestingly, we found that oral cells expressing E7 increased three times the total EGFR and pEGFR^Y1173^ levels. Moreover, amphiregulin (AREG), an EGFR ligand, was positively regulated upon E7 expression in oral cells, confirming that epithelial cells ectopically express HPV oncoproteins [29] and proposing EGFR signaling in E7-mediated effects in oral cells.

We also characterized signaling pathways involved in E7-mediated *PIR* upregulation, (using a MAPK/PI3K array) finding the activation of AKT1 and p38 in oral cells expressing HPV16 E7. In fact, AKT1^S473^, p-p70 s6 kinase, p-CREB, p-GSK3b and p-mTOR levels were significantly increased in E7-expressing oral cells; moreover, alpha, beta and gamma p38 levels were upregulated by at least 30% compared to the control cells. Likewise, E7 involvement in PI3K/AKT1 activation and was relevant during the progression of cervical cancer [48], while, regarding the EGF pathway, Gefitinib incubation did not alter Pirin levels. However, a time–response analysis showed Pirin levels decreased at 3 and 6 h after Gefitinib exposure, and in determining the function of the PI3K/AKT1 pathway, Pirin decreased progressively using the LY294002 inhibitor, proposing EGFR and PI3K/AKT1 involvement in E7-induced PIR regulation. Moreover, LY294002 inhibition of the NF-κB pathway in tumor and non-tumor oral cells demonstrates PI3K/AKT1 participation, although, constitutively, active Akt promotes κB kinase inhibitor (IKK) activation [49]. As the IKK complex is phosphorylated by AKT1, promoting nuclear NF-κB translocation (composed of c-Rel and p65^S529^), our study confirmed positive regulation by HPV E7, while the effector p65^S529^ is activated by AKT and can provide cellular survival properties [50]. Importantly, at the molecular level, NF-κB pathway transcription factors, among others, are dependent on the ability/affinity to bind with DNA; therefore, theoretically, Pirin is a coregulator of NF-κB, facilitating p65 binding to the κB gene [46]. In another study, Gustin JA et al. [51] observed that LY294002 decreased NF-κB binding to DNA in different cell models and, likewise, we demonstrated PI3K/AKT1’s potential involvement in NF-κB activation via E7, inducing *PIR* promoter transcription.

We also identified transcription factors involved in the activation of the *PIR* promoter. Two candidates were NRF2 and E2F1 transcription factors, which show an affinity for ARE- and E2F1 DNA-binding sites, respectively. In fact, the E2F transcription factor is related to E7 oncoprotein expression, pRb degradation and the release of E2F to promote cell cycle progression [52]. Alternatively, although no reports exist regarding HPV and NRF2, they are potential biomarkers for cervical cancer progression [53]. Interestingly, NRF2 is a master regulator of *PIR* promoter activity and has been raised as a possible mediator [30]. In fact, NRF2 and RNA Polymerase II were recruited to the *PIR* promoter, in the presence of E7; however, other intermediaries related to Pirin upregulation maybe involved. Previously, it has been reported that the PI3K/AKT1 pathway regulates NRF2 levels through glycogen synthase kinase 3 beta (GSK3B) phosphorylation [54,55,56] for subsequent NRF2 release from the GSK3B–CULIN3 complex, targeting it for proteasome-mediated degradation [57]. Thus, by increasing the PI3K/AKT1 pathway activation by HPV16 E7, correspondingly, NRF2 levels are expected to increase.

Finally, we found that E7-mediated *PIR* overexpression is associated with increased oral epithelial cell migration, decreased E7, p65 or PIR knockdown, effects which are similarly seen with foreskin keratinocytes [18]. In summary, our study proposes that the activation of the EGFR/PI3K/AKT1/NRF2 pathway by HPV16 E7 leads to PIR/NF-κB activation in tumor oral epithelial cells, resulting in an increased cell migration, with the possibility remaining that this mechanism is involved in other epithelial cell models (Figure 7). We believe that components of these signaling pathways constitute potential therapeutic targets of oral cancer associated to HR-HPV infection.

## 4. Materials and Methods

### 4.1. Cell Lines, Culture, Vectors, Transfections and Transductions

Floor of mouth squamous cell carcinoma (SCC143) and squamous cell carcinoma of the oral lateral tongue (SCC47) cells were obtained from the University of Pittsburgh [58]; PA317 (CRL-9078^™^) retrovirus packaging cells were obtained from ATCC^®^; keratinocyte of floor of mouth (OKF6-TERT2) and Primary Human Foreskin Keratinocytes (PHK) were kindly donated by Dr. Luisa Lina Villa (Instituto do Câncer do Estado de São Paulo, São Paulo (SP), Brazil). Cells were incubated in Dulbecco’s Modified Eagle Medium (DMEM) (Gibco, Carlsbad, CA, USA) supplemented with 10% fetal bovine serum (FBS) (Hyclone, Fremont, CA, USA) with antibiotics (100 units/mL penicillin, 100 g/mL streptomycin) and maintained at 37 °C with 5% CO_2_ atmosphere. For the subculture, cells were incubated with trypsin for 3–5 min and maintained with a new medium containing FBS (Hyclone, Fremont, CA, USA). Cells were periodically tested for mycoplasma contamination. For cell transductions, pLXSN and pLXSNHPV16E7 recombinant vectors were kindly donated by Dr. Massimo Tommasino, from the International Agency for Research on Cancer (IARC), Lyon, France. Retroviral particles were obtained from PA317 packaging cells previously transfected with pLXSN or pLXSNHPV16E7 plasmids for 24 h at 37 °C in an atmosphere containing 5% CO_2_ with lipofectamine 2000 (Invitrogen, Carlsbad, CA, USA) according to the manufacturer’s instructions. SCC143 cells were stably transduced with retroviral particles and were then selected by 0.3 mg/mL Geneticin (GIBCO, Carlsbad, CA, USA). For Pirin overexpression, pcDNA 3.1+C–eGFP and pcDNA 3.1+C-eGFP–Pirin vectors were purchased from GenScript Biotech Corp. (Piscataway, NJ, USA). 

### 4.2. Organotypic Raft Cultures

PHK and OKF6-TERT2 raft cultures were prepared as described elsewhere [59]. Raft cultures were allowed to differentiate for 14 days, the treatment with inhibitors was performed at 12 and 24 h. The rafts were then harvested, fixed in formalin and paraffin embedded.

### 4.3. Cell Viability Assay

Cells were cultured to 80–90% confluence in a 96-well flat-bottomed microtiter plate with DMEM 1× medium with 10%SFB and were treated with 0.05, 0.1, 0.25, 0.5 and 1 mg/mL Geneticin (GIBCO, Carlsbad, CA, USA) for 7 days. Finally, 20 µL MTS (Promega Corporation, Madison, WI, USA) was added to each well, the plate was incubated for 3 h at 37 °C and absorbance was measured at 492 nm using a microplate reader (BioTek Instruments, Inc., headquartered in Winooski, VT, USA).

### 4.4. Luciferase and NF-κB Activation Assays

A pmir-GLO construct containing the PIR promoter downstream from the luciferase reporter gene and mutants were purchased from GenScript Biotech Corp (Piscataway, USA). The ApE-A plasmid editor software was used (M. Wayne Davis). A reporter system was developed by using the expression vector pmiR-GLO (Promega, Madison, WI, USA). The native PGK promoter was deleted by *BglII* and *ApaI* digestion and the synthetized PIR promoter region (900 bp between 4430 and 5329 on chromosome X) by GenScript was inserted upstream of the firefly luciferase reporter gene (luc2). Renilla luciferase activity was used to normalize the firefly luciferase activity. The pHAGE NF-κB-TA-LUC-UBC-GFP-W was acquired from Addgene depositor, Darrell Kotton Lab. SCC143 cells were plated in triplicate in a 24-well plate to 70–80% confluency and were transfected with 500 ng pmiR-GLO or pHAGE luciferase reporter plasmid with 2 μL Lipofectamine 2000 (Invitrogen™, Carlsbad, CA, USA) per well. The culture medium was replaced by Opti-MEM Reduced Serum. After 18 h incubation, the transfection mixture was replaced by complete culture medium for 24 h. The cells were rinsed in 1× Phosphate Buffered Saline (PBS) and harvested with Passive Lysis Buffer in a new tube on ice. Luciferase activity was measured using the Dual-Luciferase^®^ Reporter Assay System (Promega, Madison, WI, USA), according to the manufacturer’s protocol. For the NF-κB activation assay, the culture plate was divided to measure fluorescence intensity in a flow cytometer, FACSCantoA.

### 4.5. Proteome Profiler Phospho MAPK and NF-κB Array

SCC143 cells were grown to 90% confluence in 10-cm plates and subjected to serum starvation for 24 h. Cells were collected by centrifugation and washed once with PBS. The washed cell pellets were suspended in extraction lysis buffer and incubated for 20 min at 4 °C. The protein concentration was determined by the Pierce BCA protein assay kit (Pierce, Rockford, IL, USA), according to the manufacturer’s instructions. Screening for different proteins in cell lysates was performed with the Proteome Profiler Human NF-κB Pathway Array and Proteome Profiler Human Phospho-Kinase Pathway Array (R&D Systems, Minneapolis, MN, USA). Horseradish peroxidase substrate (Millipore Corporation, Burlington, VT, USA) was used to detect the protein signal and data were captured by exposure in ChemiDoc Imaging System (Bio-Rad, Hercules, CA, USA). For semi-quantitative analysis, ImageJ software version 1.52a (National Institutes of Health, Bethesda, MD, USA) was used.

### 4.6. Western Blot

Total protein was extracted from cells with 1× RIPA lysis buffer (Abcam, Cambridge, UK) containing both protease and phosphatase inhibitor cocktail (Roche, Basel, Switzerland). Cells were incubated at 4 °C for 10 min, sonicated on ice for 20 s and centrifuged at 14,000× *g* for 15 min. The proteins were quantified by using the Pierce BCA Protein assay kit (Pierce, Rockford, IL, USA) and 30 µg of the extract was loaded per well. Following 12% PAGE)–SDS, the proteins were transferred to a Hybond-P ECL membrane (Amersham, Piscataway, NJ, USA) using 20 mM Tris, 150 mM glycine pH 8.3, in 20% methanol with the semi-dry transfer apparatus (Bio-Rad, Hercules, CA, USA). Membranes were incubated for 1 h at room temperature with blocking buffer (5% bovine serum albumin, Tris-buffered saline (TBS)–0.5% Tween 20, pH 7.6) and incubated overnight at 4 °C with primary antibody against Pirin (ab51360), pRb (ab24), EGFR (ab32562), EGFR^Y1173^ (ab32578), EGFR^Y1068^ (ab40815), p65^s536^ (ab86299), β-actin (ab6276) (Abcam, Cambridge, UK), C-Rel (MAB2699), p65^s529^ (MAB7624) (R&D System, Minneapolis, MN, USA), HPV16 E7 (SC6981), pAKT1-2-3 (SC514032), PHOSPHO AKT^S473^ (4060S), NRF2 (4060S) (Cell Signaling, Danver, MA, USA), ERK (SC514302), pERK (SC136521), AREG (SC5797), pMTORC (SC293133), MTORC (SC517464), Histone H3 (SC56616), GSK3 (SC7291), p65 (SC8008), (Santa Cruz Biotechnology, Dallas, TX, USA) diluted 1/1000 in Tris-buffered saline–Tween 20 (TBS–T20). Membranes were washed three times in TBS–T20 and incubated with secondary anti-IgG-labeled peroxidase (Anti Mouse 554002; BD Pharmingen; BD Biosciences and Anti Rabbit SC2004; Santa Cruz Biotechnology, Inc., Dallas, TX, USA). After three washes in TBS–T20, immune complexes were detected using the ECL system (Amersham Pharmacia Biotech, Little Chalfon, UK) according to the manufacturer’s instructions. Detailed information can be found at Appendix A.

### 4.7. Pharmacological Inhibition Assay

For inhibition experiments, SCC143 cells were serum-depleted for 24 h and incubated with 100 µM Gefitinib (EGFR inhibitor) (Sigma, St. Louis, MO, USA), 6.3 µM UO126 (ERK inhibitor) (Sigma, St. Luis, MO, USA) or 10 µM LY294002 (PI3K inhibitor) (Sigma, St. Luis, MO, USA) for times ranging from 1.5 to 24 h. Then, cells were washed twice with 1× PBS (pH 7.4), dried and lysed with 1× RIPA buffer (Abcam, Cambridge, UK) containing both protease and phosphatase inhibitor cocktail (Roche, Basel, Switzerland).

### 4.8. Immunofluorescence and Confocal Microscopy

The cellular location of Pirin and E7 proteins was determined by using indirect immunofluorescence (IF). Transduced cells were grown to confluence in Chamber Slides, washed twice with 1× PBS (pH 7.4), dried and incubated for 5 min with cold Paraformaldehyde (PFA). Next, cells were incubated with 3% bovine seroalbumin (BSA) for 1 h at room temperature, followed by incubation with a primary monoclonal anti-specific protein antibody diluted in 1× PBS (1:50), according to the manufacturer’s instructions. The fixed cells were washed three times for 5 min at room temperature and incubated with a secondary fluorescein isothiocyanate (FITC) or Texas red-labeled anti-IgG antibody (1:100) (Santa Cruz biotechnology, Dallas, TX, USA). After three washes with 1× PBS, cells were incubated for 15 min with Rhodamine Phalloidin (Rh/Ph) and DAPI (Thermo Fisher Scientific, Waltham, MA, USA), and finally visualized in a C2 Plus confocal microscope.

### 4.9. Reverse Transcriptase–Quantitative Polymerase Chain Reaction

RNA from cell lines was isolated using Trizol reagent (Invitrogen, Carlsbad, CA, USA) according to the manufacturer’s instructions. After chloroform purification and isopropanol precipitation, the RNA was suspended in diethylpyrocarbonate (DEPC)-treated water and stored at −80 °C. Next, the RNA was treated with RQ1 RNase-free DNAse (Promega, Madison, WI, USA) at 37 °C for 60 min and then incubated with RQ1 DNAse Stop Solution for 10 min. cDNA was prepared using a 20 µL-reaction volume containing DNAse treated RNA (2 µg), 1 U RNAse inhibitor (Promega, Madison, WI, USA), 0.04 µg/µL random primers (Promega, Madison, WI, USA), 2 mM dNTPs (Promega, Madison, WI, USA) and 10 U Moloney Murine Leukemia Virus (MMLV) reverse transcriptase (Promega, Madison, WI, USA). The reaction mixture was incubated for 1 h at 37 °C. cDNA was subjected to amplification by real-time quantitative PCR using specific primers for transcripts (Table 1) in a Rotor-Gene 6000 apparatus (Corbett Research, Sydney, Australia). The conditions of amplification were as follows: 95 °C for 10 min followed by 40 cycles of denaturation at 95 °C for 15 s, annealing at 55 °C for 20 s and extension at 72 °C for 20 s. The dissociation curve was carried out by increasing the temperature from 70 to 90 °C, by 0.5 °C at each step. The reaction was performed using 2 × SYBR Green Master Mix (Bioline, London, UK), 0.4 µM primers, 10.5 µL RNAse-free water and 1 µL cDNA in a 25 µL final volume. Specific primers and amplification conditions were adjusted for each RNA sequence. The relative copy number of each sample was calculated by the 2^(−ΔΔ*C*(T))^ method and Rotor-Gene software. Endogenous β-actin mRNA levels were used for the normalization of RNA expression. All reactions were performed in duplicate.

### 4.10. Cell Migration Assays

For the three-dimensional migration assay, the bottom side of a transwell upper chamber (Corning, New York, NY, USA) was coated with 2 µg mL^−1^ fibronectin and incubated overnight at 4 °C. A total of 15,000 SCC143 cells were seeded in non-supplemented medium inside the upper chamber, and 500 µL of complete medium was added to the plate. Cells were allowed to migrate for 7 h. Migrated cells were fixed and stained with crystal violet/methanol solution. Non-migrated cells were removed with a cotton tip. Migrated cells were counted in eight fields for each experiment.

### 4.11. Chromatin Immunoprecipitation

SCC143 cells stably transfected with pLXSN and pLXSNE7 were grown in 10 cm dishes until reaching 80% confluence. SCC143-E7 cells were treated with 10 μM LY294002 for 12 h and were subsequently incubated with 1% formaldehyde at room temperature to elicit DNA-protein cross-linking. DMSO-treated cells were included as controls. After 2 min of incubation, 0.4 M glycine was added to quench the reaction. Next, the cross-linked cells were lysed with a scraper and harvested by centrifugation at 8000 rpm for 20 min at room temperature. After centrifugation, supernatants were removed, pellets were washed twice in Triton X-100 buffer (50 mM Tris, pH 7.4; 150 mM NaCl; 1 mM EDTA; 0.1% Triton X-100; 0.1% Nonidet P-40; 0.01% SDS) and subjected to three sonication pulses (1 min each/pulse). Afterwards, the volume of sonicated chromatin was adjusted to 4 mL with Triton buffer–SDS 0.1%, then cleared by centrifugation at 13,000 rpm for 10 min. Chromatin was stored in 800-μL aliquots at –80 °C. Immunoprecipitation of NRF2, Pol II, c-JunS73 and c-FosT232 bound to chromatin was carried out by using a mixture composed by 80 μL of pre-blocked immunoglobulin G (IgG)-magnetic beads, an equal volume of chromatin dissolved in Triton buffer and 3 μL of specific antibody. Before immunoprecipitation with IgG beads, 10% of each sample was saved as an input fraction. The mixture was incubated overnight at 4 °C in a rotary shaker. The next day, the beads were pelleted by centrifugation at 12,000 RPM for 1 min. Five hundred μL of supernatant was washed four times in 400 μL of Triton buffer and once with PBS buffer pH 7.4. After each wash, the liquid suspensions were transferred to fresh tubes. Immunocomplexes were de-cross-linked by overnight incubation at 85 °C in 125 μL of elution buffer (50 mM Tris-HCl, pH 8.0, 10 mM EDTA, 1% SDS, 2.5% dithiothreitol). Free DNA was purified from the solution by phenol–chloroform extraction. After precipitation by the addition of 3M Sodium acetate and cold ethanol, DNA was washed in 70% ethanol and dried for 20 min using a Speedvac Rotation System. Finally, purified DNA was suspended in 12 μL of Tris-EDTA (TE) buffer (10 mM Tris-Cl; 1 mM EDTA) for PCR analysis. PCR amplifications were performed using a RotorGene 6000 system with an initial hold at 94 °C for 5 min, followed by 35 cycles of 1 min at 94 °C, 1 min at 58 °C, and 2 min at 72 °C and then one cycle of 4 min at 72 °C. Primers that span the antioxidant response elements (ARE) were used for qPCR analysis.

### 4.12. Statistical Analysis

A comparison of means between two groups was performed using a Mann–Whitney test, whereas comparisons between multiple groups were performed using one-way ANOVA and Tukey’s post-hoc test. Data were analyzed using GraphPad Prism version 7 software (GraphPad Software, Inc., San Diego, CA, USA). *p*-values of less than 0.05 were considered statistically significant.

## 5. Conclusions

Our data allow us to propose that activation of the EGFR/PI3K/AKT1/NRF2 pathway by HPV16 E7 leads to PIR/NF-κB activation in oral tumor epithelial cells, finally leading to an increased cell migration, although a possibility still exists that this mechanism is involved in other epithelial cell models. It is tempting to speculate that components of these signaling pathways may constitute therapeutic targets of oral cancer associated to HR-HPV infection.

## Figures and Tables

**Figure 1 cancers-12-01904-f001:**
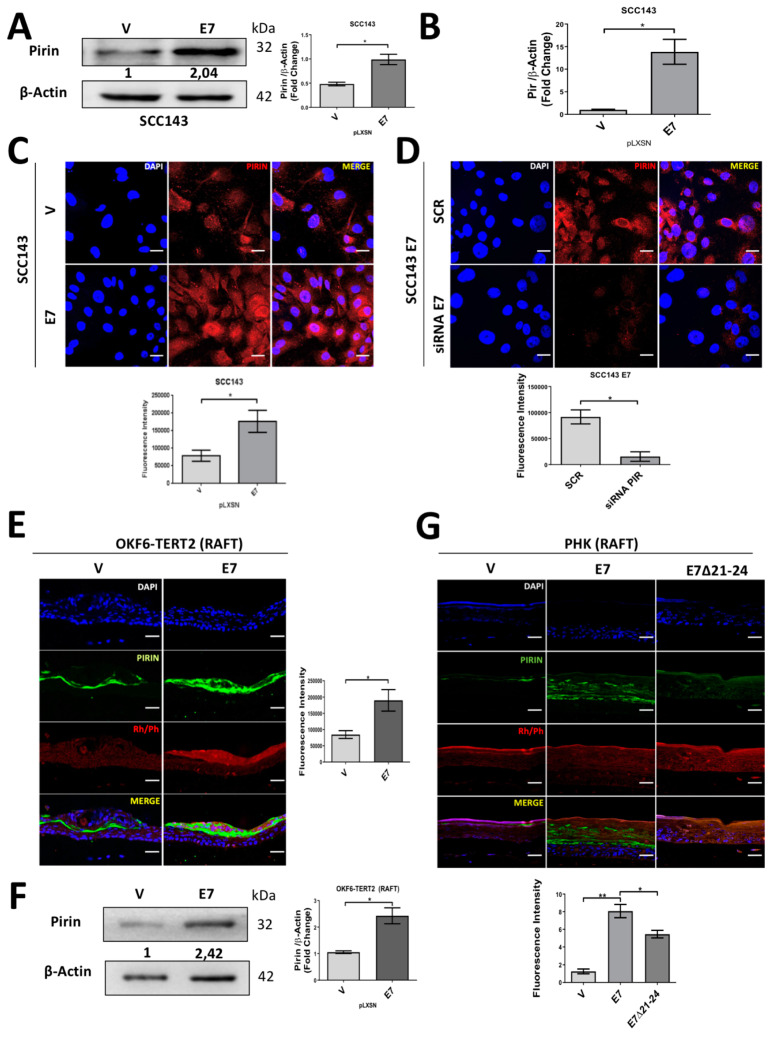
Human papilloma virus (HPV)16 E7 oncoprotein positively regulates the levels of Pirin protein in oral cells and foreskin keratinocytes. (**A**) Western blot to evaluate Pirin protein levels in SCC143/E7 and SCC143/V cells using β-actin as a loading control. The graphs represent a densitometric analysis of three independent Western blots for each protein normalized by β-actin. (**B**) RT-qPCR was performed for the normalized PIR transcript with the β-actin transcript. (**C**) Indirect immunofluorescence (IFI) reveals an increase in Pirin levels in SCC143/E7 cells. Scale bar: 10 µm. (**D**) IFI performed on SCC143/E7 cells previously transfected with control siRNA (SCR) and E7 siRNA to evaluate Pirin protein. Scale bar: 10 µm. (**E**) IFI performed on OKF6/TERT2 V and E7 oral organotypic raft culture cells to evaluate Pirin protein. Scale bar: 35 µm. (**F**) Western blot to evaluate Pirin protein levels in organotypic raft cultures established from OKF6/TERT2 V and OKF6/TERT2 E7 oral cells. (**G**) IFI performed in human foreskin keratinocytes (HFK)-expressing E7, E7Δ21-24 and the control with the empty vector; graph represents fluorescence analysis. Scale bar: 35 µm. Data are presented as the mean ± standard error of the mean (SEM); average of three independent experiments, conducted in triplicate. * *p* < 0.05 and ** *p* < 0.01 (Mann–Whitney test). Rhodamine Phalloidin (Rh/Ph).

**Figure 2 cancers-12-01904-f002:**
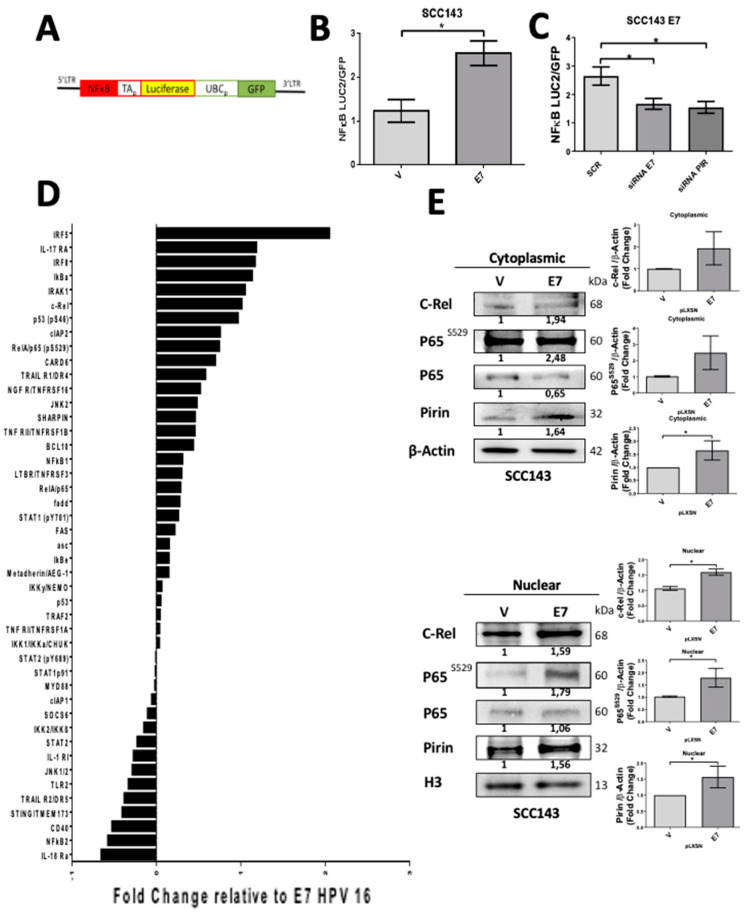
HPV16 HPV E7 promotes nuclear factor kappa-light-chain-enhancer of activated B cell (NF-κB) activation in oral cells. (**A**) pHAGE/NF-κB reporter vector map. (**B**) Luciferase activity normalized with GFP was evaluated in SCC143/E7 and SCC143/V cells transfected with the reporter vector pHAGE/NF-κB. (**C**) Luciferase activity normalized with GFP in SCC143/E7 cells co-transfected with pHAGE/NF-κB and siRNA for PIR or siRNA E7 knockdown and siRNA (SCR) as a control. (**D**) Protein array of NF-κB signaling pathway comparing the SCC143/V and SCC143/E7 cell extracts. Data were plotted in reference to the change that occurred in the presence of E7 compared to the empty vector control. (**E**) Western blot of nuclear and cytoplasmic protein fractions was performed to analyze the levels of Pirin, p65^S529^, p65, and C-Rel. β-actin or H3 were used as a load control in SCC143 cells transduced with empty (pLXSN) or E7 constructs. The graphs represent a densitometric analysis of three independent Western blots (WBs) for each protein normalized against β-actin. Data are presented as the mean ± SEM; average of three independent experiments, conducted in triplicate. * *p* < 0.05 (Mann–Whitney test).

**Figure 3 cancers-12-01904-f003:**
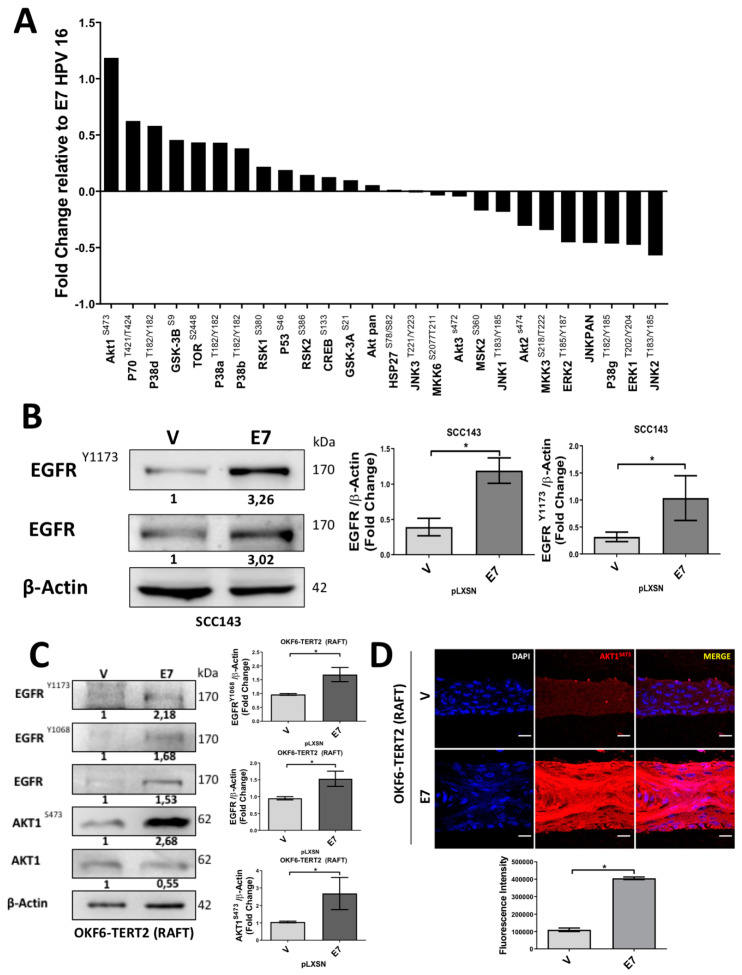
HPV16 HPV E7 protein induces EGFR/PI3K/AKT signaling. (**A**) Protein array of the MAPK and AKT pathways comparing the SCC143/V and SCC143/E7 cell extracts. Data were plotted in reference to the change that occurred in the presence of E7 compared to the empty vector control. (**B**) Western blot performed to check the levels of total EGFR, pEGFR^Y1173^ and β-actin used as load control in SSC143/E7 and SSC143/V cells. The graphs represent a densitometric analysis of three independent assays. (**C**). WBs were performed with protein extracts from OKF6-TERT2 cell organotypic cultures, the levels of total EGFR, pEGFR^Y1173^, pEGFR^Y1068^, AKT1, pAKT1^S473^, pRb, Pirin and β-actin, used as load controls, were analyzed. (**D**) IFI reveals an increase in AKT1^S473^ in OKF6-TERT2 E7 organotypic cultures. The graph represents the fluorescence analysis. Scale bar: 15 µm. Data are presented as the mean ± SEM; average of three independent experiments, conducted in triplicate. * *p* < 0.05 (Mann–Whitney test).

**Figure 4 cancers-12-01904-f004:**
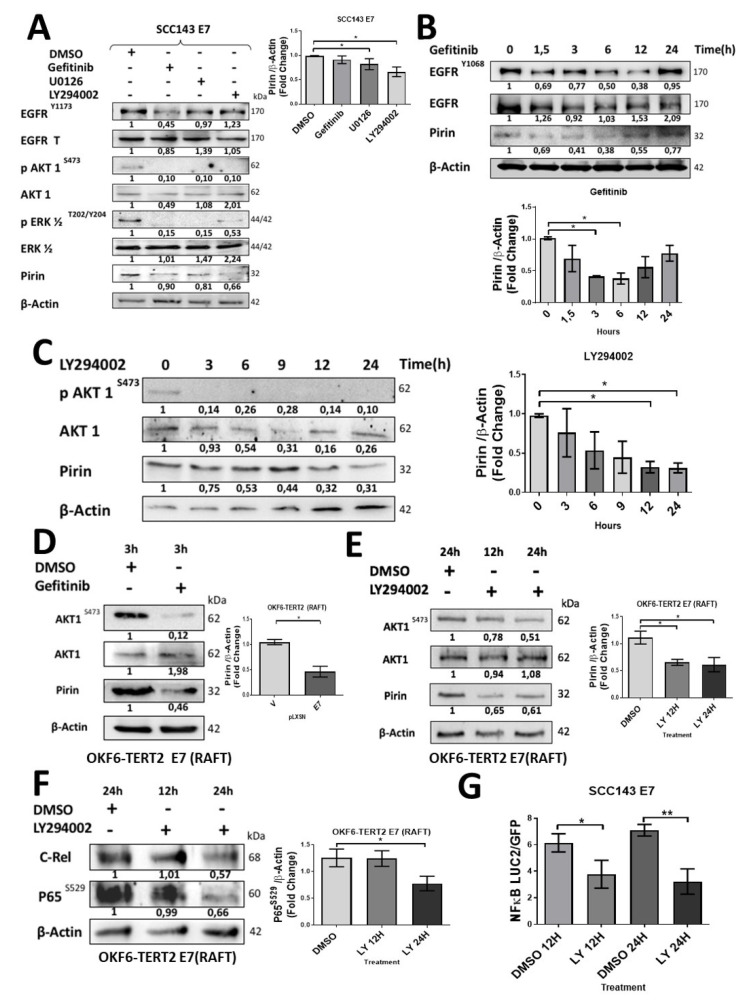
HPV16 HPV E7 induces EGFR/PI3K/AKT1 signaling for PIR/NF-kB activation in oral cells. (**A**) Western blot was performed with protein extracts from SCC143/E7 cells previously exposed to Gefitinib (EGFR), U0126 (ERK) and LY294002 (PI3K) inhibitors for 24 h. The levels of total EGFR, pEGFR^Y1173^, AKT1, pAKT1, ERK, pERK, Pirin and β-actin used as load control were analyzed. The graphs represent a densitometric analysis of three independent WBs for Pirin normalized against β-actin. (**B**) Time–response assay by exposure to Gefitinib for 1.5 to 24 h in SCC143/E7 cells. The levels of total EGFR, pEGFR^Y1068^, Pirin and β-actin used as load control were analyzed. The graph represents a densitometric analysis of three independent experiments. (**C**) Time–response assay by exposure to 10 µM LY294002 for 3 to 24 h in SCC143/E7 cells. The levels of total AKT1, pAKT1, Pirin and β-actin used as load control were analyzed. The graph represents densitometric analysis of three independent experiments. (**D**) Western blot to evaluate AKT1, pAKT1, Pirin protein levels in organotypic raft cultures established from OKF6/TERT2 E7 oral cells treated with dimethyl sulfoxide (DMSO) or Gefitinib for 3 h (β-actin used as load control were analyzed). (**E**) Western blot to evaluate AKT1, pAKT1, Pirin protein levels in OKF6/TERT2 E7 oral organotypic raft culture cells treated with DMSO or 10 µM LY294002 for 12 or 24 h (β-actin used as load control were analyzed). (**F**) Western blot to evaluate c-Rel and p65^S529^ protein levels in organotypic raft cultures established from OKF6/TERT2 E7 oral cells treated with DMSO or 10 µM LY294002 for 12 or 24 h (β-actin used as load control were analyzed) (**G**) Luciferase activity normalized against GFP was evaluated in SCC143/E7 cells transfected with the reporter vector pHAGE/NF-κB treated with DMSO or 10 µM LY294002 for 12 or 24 h. Data are presented as the mean ± SEM; average of three independent experiments, conducted in triplicate. * *p* < 0.05 and ** *p* < 0.01 (ANOVA test). LY294002 (LY).

**Figure 5 cancers-12-01904-f005:**
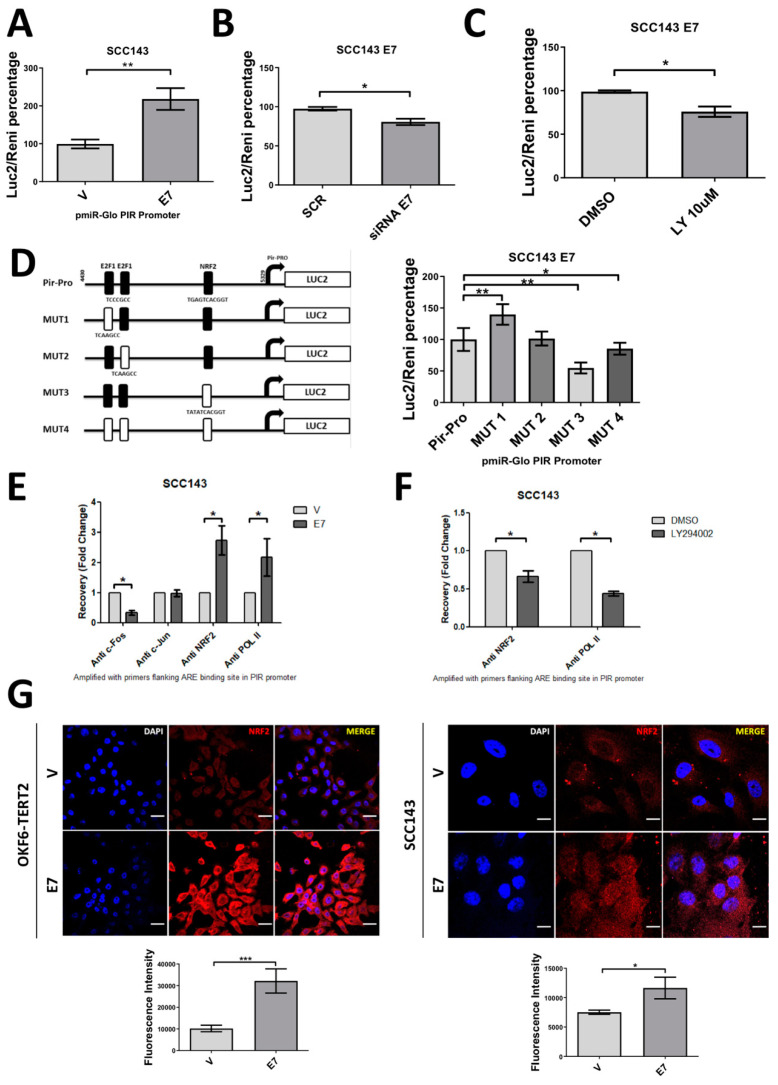
HPV16 HPV E7 promotes PIR-Promoter activation through NRF2 transcription factor in oral cells. (**A**) Luciferase activity was measured in SCC143/V and SCC143/E7 cells previously transfected with pmiR-GLO/pPIR vector for 18 h (Renilla activity was used as a normalizer). (**B**) Luciferase activity was measured in SCC143/E7 cells previously co-transfected with pmiR-GLO/pPIR vector and a scrambled siRNA (SCR) or siRNA E7. Luciferase activity was measured 18 h post-transfection. (**C**) Luciferase activity was measured in SCC143/E7 cells previously transfected with pmiR-GLO/pPIR vector and treated with 10 µM LY294002 for 12 h (Renilla activity was used as normalizer). (**D**) The PIR promoter was mutated as indicated in the map; each mutant was cloned into the pmiR-GLO vector and designated as 1, 2, 3 and 4. Luciferase activity was measured in each of the mutants in SCC143/E7 cells (Renilla activity was used as normalizer). (**E**) ChIP assay performed in SCC143/E7 and SCC143/V cells, precipitation was performed by using c-Jun, c-Fos, NRF2 and RNA Pol II antibodies. (**F**) ChIP performed in SCC143/E7 cells treated with DMSO or 10 µM LY294002 for 12 h; precipitation was performed with NRF2 and RNA Pol II antibodies. (**G**) IFI reveals the increase in NRF2 in oral cells expressing HPV16 E7. Scale bar: 10 µm for OKF6-TERT2 cells and 5 µm for SCC143 cells. Data are presented as the mean ± SEM; average of three independent experiments, conducted in triplicate. * *p* < 0.05 and ** *p* < 0.01 (ANOVA test).

**Figure 6 cancers-12-01904-f006:**
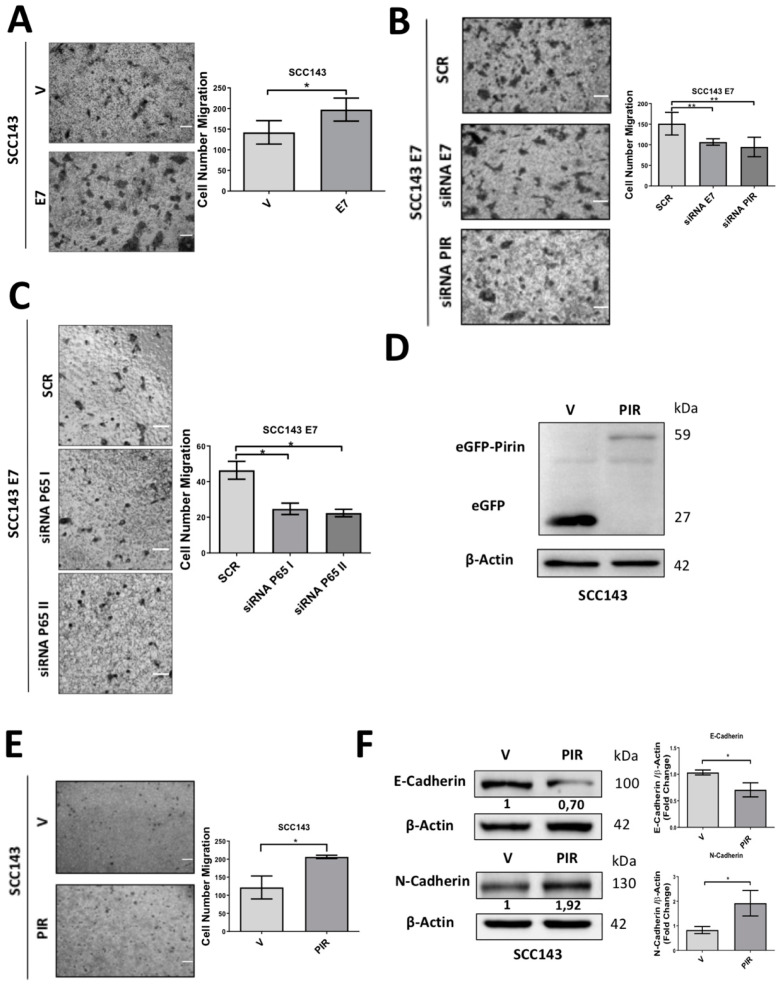
HPV16 E7 expression and PIR induces migration of oral cells. (**A**) A migration assay in SCC143/E7 and SCC143/V cells was carried out for 7 h using fibronectin pretreated transwells, scale bar 25 µm. (**B**) A migration assay performed on SCC143/E7 cells previously transfected with control siRNA (SCR), siRNA PIR or HPV16 siRNA E7 was carried out for 7 h using fibronectin pretreated transwells, scale bar 25 µm. (**C**) A migration assay performed on SCC143/E7 cells previously transfected with control siRNA (SCR) and siRNA I-II p65 was carried out for 7 h using fibronectin pretreated transwells, scale bar 25 µm. (**D**) Western blot against eGFP performed for V corresponding to the empty vector (pcDNA 3.1–eGFP) and PIR corresponding to the vector containing the PIR sequence linked to eGFP (pcDNA 3.1–eGFP–PIR) transfected in SCC143 oral cell. (**E**) A migration assay in SCC143 V (pcDNA 3.1–eGFP) and SCC143 PIR (pcDNA 3.1–eGFP–PIR) cells was carried out for 7 h using fibronectin pretreated transwells. Scale bar: 40 µm. (**F**) Analysis of E-cadherin and N-cadherin protein levels, which were normalized with the expression of β-actin. The graphs represent a densitometric analysis of three independent assays. Data are presented as the mean ± SEM; average of three independent experiments, conducted in triplicate. * *p* < 0.05 and ** *p* < 0.01 (Mann–Whitney test).

**Figure 7 cancers-12-01904-f007:**
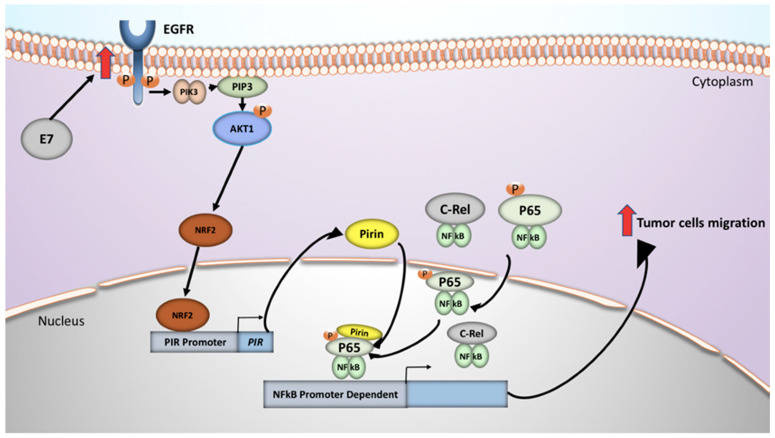
A proposed model of signaling pathways involved in HPV16 E7-mediated Pirin upregulation in oral epithelial cells.

**Table 1 cancers-12-01904-t001:** Primers used in this study

Region	Forward 5′-3′	Reverse 5′-3′	Amplicon (bp)
PIR	TCAAATTGGACCCAGGAGCC	TCCAAGCACTGCTGTGTGAT	131
E7 HPV16	CAATATTGTAATGGGCTCTGTCC	ATTTGCAACCAGAGACAACTGAT	120
ARE CHIP	GTGAGCATCTCTTCCCGGCA	TGACCGCCAGCATTCCCTCA	347
β-actin	AGCGAGCATCCCCCAAAG	GGGCACGAAGGCTCATCA	289

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
