# Peer review of "Human Papillomavirus 16 E7 Promotes EGFR/PI3K/AKT1/NRF2 Signaling Pathway Contributing to PIR/NF-κB Activation in Oral Cancer Cells"

_cancers, 2020, doi:10.3390/cancers12071904_

Round 1
Reviewer 1 Report
All concerns have been addressed adequately.
Author Response
We appreciate the reviewer´s comments and criticisms.
Reviewer 2 Report
The authors have addressed most of the concerns from the previous round of review, but the following still should be addressed:
- The authors state in the materials and methods where they obtained SCC143 cells, but they do not tell the reader what kind of cells they are: cancer or not? Tissue of origin?
- The labels on the bar graphs are still too small to be legible.
Author Response
We appreciate the reviewer´s comments.
- The authors state in the materials and methods where they obtained SCC143 cells, but they do not tell the reader what kind of cells they are: cancer or not? Tissue of origin?
ANSWER. A sentence was added in Material and methods. A143 cells were originated from a floor of mouth squamous cell carcinoma
2. The labels on the bar graphs are still too small to be legible.
ANSWER. All the graphs were corrected.
This manuscript is a resubmission of an earlier submission. The following is a list of the peer review reports and author responses from that submission.
Round 1
Reviewer 1 Report
The authors describe in the manuscript that the HPV16 E7 protein activates EGFR/PI3K/AKT1/NRF2 pathways to induce NF-kappaB signaling via pirin expression. The paper is a follow-up to PMID:29118270 from the same group. This paper already described that HPV16 E7 induces pirin via EGFR/PI3K/AKT1 pathways in cervical and oral epithelial cells. In the present manuscript, the authors use OKF6-Tert2 cells, which were already used in the previous paper, and HPV-negative SCC143 cancer cells. in which they express HPV16 E7. Confirmatory of their published work they find that pirin is induced. Using protein arrays they observe that NF-kappa B, AKT1 and EGFR pathways are disturbed by E7. Using siRNAs and pathway inhibitors they come to the conclusion that E7 activates an EGFR/AKT1/NRF2 pathway to induce pirin which in turn enhances p65, a NF-kappa B member, which results in tumor cell migration.
Overall, the novel information provided by this manuscript compared to the previous paper is rather small. The role of NRF2 in the activation of pirin expression and the functional relevance of p65/relA induction are not sufficiently mechanistically explored to warrant a publication in Cancers at this point.
Major points:
1) There is strong epidemiological and genetic evidence that HPV-negative and –positive oral cancers are very different entities. Instead of using HPV-negative SCC143 cells and making them transgenic for E7, the authors should use HPV-positive HNSCC cell lines in order to mimic the in vivo situation.
2) Many of the graphs are far too small. Even at high magnification in the PDF reader, the labelings cannot be read.
3) The labeling of the x-axis in Figure 2D and of the y-axis in Fig. 3A is fold-change relative to E7. Please explain what negative changes are. Why are there no error bars? The majority of changes are rather small. Are they significant? Please comment and explain.
4) Figure 1E,G: What is Rh/Ph?
5) Figure 1A,B. The changes at the pirin protein level are much weaker than at the RNA level. Please comment.
6) Figure 3D: The AKT1-S473 staining looks weird. Please provide IF data for total AKT1 to rule out non-specific staining or an artifact.
7) Figure 5B: The effect of the E7 knock-down on PIR promoter is very small. How efficient is the knock-down? Figure 5E, F: No error bars or data points are provided. How often was the experiment repeated?
8) Figure 7: what is the evidence that p65 contributes to tumor cell migration?
Reviewer 2 Report
In addition to its activity to promote degradation of pRb, E7 is able to mediate a variety of other changes in cellular signaling and behavior. In this report, Carrillo-Beltran et al describe a pathway in which E7 promotes increased EGFR activity, resulting in signaling that induces expression of Pirin, which, in turn, helps activate NFkB. This paper is well constructed and reasonably strong, but a few weaknesses should be addressed.
The labels on the figures are far too small, especially the bar graphs, which are essentially unreadable. The authors should specify what SCC143 cells are. If the cells are derived from a cancer, then some of the signaling and migration behavior may already be abnormal in the absence of E7. Accordingly, data in Figure 6 show relatively modest differences between E7-expressing vs control cells. Could this be due to a high rate of migration, even in the controls? The authors should repeat these experiments using E7-expressing PHKs, in which the differences may be more pronounced. The proteomics data are confusing in several ways. The scanned profiler array blots are shown, which is good, but the spots that are discussed in the text should be labeled. In both array experiments, the visual impression is that E7 largely reduces spot intensity, whereas the authors discuss a large number of factors increasing. Can the authors explain this disparity? How many times were these experiments repeated? In Figure 2 and Figure 3, a number of pathways are said to be upregulated or downregulated. The authors only choose to focus on the upregulated ones, even though the downregulated pathways seem just as relevant. Why is that? The authors should make clear that total p65 levels do not change, and that nuclear p65 levels increase in the presence of E7, and that p65S529 refers to phosphorylation, which promotes nuclear translocation. The present treatment in the text is confusing. After having implicated PI3K/AKT signaling in PIR gene expression, they do promoter analysis focusing on E2F and ARE elements. This does not seem to follow. The authors should explain their reasoning for making those mutants. Line 224 – It appears that Figure 3C should actually be 5D. Figures 5e and f do not have any statistics. How many times were these experiments done? Chip experiments are notoriously variable, so how many replicates are represented should be clearly shown.